# Photonic Sintering of Oxide Ceramic Films: Effect of Colored $Fe_xO_y$ Nanoparticle Pigments

**Evgeniia Gilshtein** [1,*], **Stefan Pfeiffer** [2], **Severin Siegrist** [1], **Vitor Vlnieska** [1], **Thomas Graule** [2] **and Yaroslav E. Romanyuk** [1]

[1] Laboratory for Thin Films and Photovoltaics, Empa, Swiss Federal Laboratories for Materials Science and Technology, Überlandstrasse 129, 8600 Dübendorf, Switzerland; severin.siegrist@empa.ch (S.S.); vitor.vlnieska@empa.ch (V.V.); yaroslav.romanyuk@empa.ch (Y.E.R.)

[2] Laboratory for High Performance Ceramics, Empa, Swiss Federal Laboratories for Materials Science and Technology, Überlandstrasse 129, 8600 Dübendorf, Switzerland; stefan.pfeiffer@empa.ch (S.P.); thomas.graule@empa.ch (T.G.)

\* Correspondence: evgeniia.gilshtein@empa.ch

**Abstract:** Alumina and zirconia thin films modified with colored nano-$Fe_xO_y$ pigments were sintered by the flash-lamp-annealing method. We selected a nano $\alpha$-$Al_2O_3$ and micron $\alpha$-$Al_2O_3$ bimodal mixture as the base precursor material, and we doped it with 5 vol% of $Fe_xO_y$ red/brown/black/yellow pigments. The coatings were deposited from nanoparticle dispersions both on glass and on flexible metal foil. The characteristics of the thin films obtained with the use of various additives were compared, including the surface morphologies, optical properties, crystallinities, and structures. Flash lamp annealing was applied with the maximum total energy density of 130 J/cm$^2$ and an overall annealing time of 7 s. Based on the simulated temperature profiles and electron-microscopy results, a maximum annealing temperature of 1850 °C was reached for the red $Al_2O_3$: $Fe_2O_3$ ceramic film. The results show that red $\alpha$-$Fe_2O_3$ pigments allow for the achievement of maximum layer absorption, which is effective for flash lamp sintering. It was also possible to use the selected red $\alpha$-$Fe_2O_3$ particles for the flash-lamp-assisted sintering of $ZrO_2$ on a 30 μm-thin flexible stainless-steel substrate.

**Keywords:** ceramic coating; flash lamp annealing; photonic sintering; iron oxide nanoparticle pigment

## 1. Introduction

A heat-treatment process (including post-annealing) is generally required to obtain sintered ceramic films with the desired properties, such as high wear resistance, chemical- and environmental-corrosion resistance, and high hardness [1–3]. These sintering processes are performed at temperatures above 1400 °C in conventional furnaces under controlled environments, which results in long overall fabrication times [4]. Furthermore, exposure to a high-temperature environment for several hours limits the choice of suitable substrate materials. The $Al_2O_3$ coatings with superior electrical and wear insulation are prepared using the typical methods, including the thermal spray, sol−gel, electrophoretic, and sputtering methods, as well as plasma electrolytic and electrolytic deposition in aqueous solutions [5–7]. However, often poor bonding, easy cracking, thickness limitation, or the high cost of the abovementioned methods limit the coating application. Many alternative heat-treatment methods have been investigated, including excimer lasers, microwaves, and arc plasma methods, to overcome the disadvantages of the conventional thermal-sintering process [8–11]. Laser powder bed fusion is a technique that uses a high-power laser to fuse small particles of plastic, metal, or ceramic. However, the small spot size and single wavelength of the light, and most importantly, the high thermal stresses generated in the material, impede the widespread application of the technique. In microwave sintering

for high-temperature applications, a thermal runaway develops in the sample and causes thermal instability during the sintering process.

Flash lamp annealing (FLA), or photonic sintering/curing (intense pulsed light), is one of the methods of thermal processing that utilizes ultrashort (0.1–10 ms) pulses from a xenon flash lamp (with a broadband of a 200–1000 nm spectrum) [12]. The rapid transient heating of surfaces illuminated by the lamp offers extremely fast heating rates [13,14]. Because heat is generated within the surface/layer, this method allows the annealing of thin films, even on the temperature-sensitive substrate, without its damage.

The FLA-assisted sintering of only a few ceramics, such as scandia-stabilized zirconia films [15], yttria-stabilized zirconia films [16], hafnia-zirconia thin films [17], and titanium dioxide samples [18], has been demonstrated. However, the photonic processing of alumina is challenging due to its low absorption in the visible range because of a wide bandgap of 7–8 eV. We have demonstrated previously that, by mixing the precursor alumina particles with reddish-brown-colored iron oxide nanoparticles that were used initially for selective laser sintering [19,20], it was possible to boost the optical absorption of alumina precursors [21]. Two questions, however, remained unanswered: (i) What pigments are the most effective for sintering alumina? and (ii) Can this approach be extended to other ceramic precursors? We intend to address these questions in the present study by admixing various pigments with red, brown, yellow, or black colors with the alumina dispersions. Commercially available pigments composed of metal oxides [22] with maximum light absorption in the different wavelength regions were used for these purposes. The main novelty is that the interaction between a high-power flashlight and light-reflecting white alumina can be even more enhanced by adding the homogeneous doping of red-colored iron oxide nanoparticles. We identified red-colored $Fe_2O_3$ as the most efficient pigment, and we then applied it to zirconia, which is another ceramic material that is widely used for biomedical applications. We finally demonstrated FLA-sintered nanometer-thin ceramic zirconia coatings on a flexible metal foil, which fully withstood the annealing process.

## 2. Materials and Methods

To obtain thin and uniform layers, we utilized a bimodal mixture of the $\alpha$-$Al_2O_3$ with an average particle size of 3 μm (micrometer $Al_2O_3$), submicron $\alpha$-$Al_2O_3$ with a size of 200 nm (nanometer $Al_2O_3$), and the doping of 5 vol% $Fe_xO_y$ red/brown/black/yellow pigments in DI water. Submicron $\alpha$-$Al_2O_3$ Taimicron TM-DAR (Taimei Chemicals Co. LTD) and micron $\alpha$-$Al_2O_3$ AA3 (Sumitomo, Chemical Co. LTD) were purchased as ceramic raw materials. Nano-$Fe_xO_y$ additives (Bayferrox) were used as coloring agents (pigments). These included: iron oxide $\alpha$-$Fe_2O_3$ Bayferrox Red 110 in the form of spherical nanoparticles with a 90 nm average particle size; iron oxide $Fe_3O_4$ Bayferrox Black 330 spherical nanoparticles with a 150 nm average size; iron and manganese oxide $(Fe, Mn)_2O_3$ Bayferrox Brown 645 T spherical nanoparticles with a 300 nm average size; iron hydroxide $\alpha$-$FeOOH$ Bayferrox Yellow 415 prismatic nanoparticles with a 200 nm × 300 nm average size. $ZrO_2$ nanoparticles (TZ-0), with an average particle size of 40 nm, were purchased from Tosoh (Tosoh-Zirconia).

**Particle dispersion.** Ammonium citrate dibasic (98%, Sigma Aldrich Corp., St. Louis, MO, USA) was used as a surfactant to achieve a homogeneous dispersion of the particles in DI water. This dispersant was previously proven to be suitable for $Fe_2O_3$ and $Al_2O_3$ ceramic particles dispersed in water [23]. The $Fe_xO_y$ pigments were all initially dispersed using vibration milling for 20 min, with a vibrational frequency of 30 Hz (Retsch MM301, Retsch GmbH, Haan, Germany). All the powders ($Fe_xO_y$ and $Al_2O_3$) were separately mixed with milling balls (1 mm diameter for TM-DAR, 5 mm diameter for AA3, and 0.5 mm diameter for $Fe_xO_y$) in three different bottles, with dispersing agent where needed, and DI water. Later, all the components were mixed to obtain a three-modal mixture, and an additional roll-milling step was applied for 24 h. A bimodal distribution of alumina particles was chosen in accordance with [20,23]. Additionally, 5 vol% of $Fe_xO_y$ pigments with respect to the entire inorganic content was added to the dispersion. Photonic-sintering experiments

were carried out for the coatings obtained with different amounts of the $Fe_xO_y$ dopant; however, the concentration of nanoparticle pigments below 5 vol% was not sufficient for FLA-assisted sintering. Figure S1a gives the zeta potentials of $Fe_xO_y$ pigments dispersed in water as a function of the pH. The zeta-potential analysis was performed to obtain information about the surface electrostatic charges of $\alpha$-$Fe_2O_3$, $Fe_3O_4$, (Fe, Mn)$_2O_3$, and $\alpha$-FeOOH with the pH variation, and to estimate the amount of ammonium citrate to be added to the nanoparticle solutions. Ammonium citrate dibasic in the case of red $\alpha$-$Fe_2O_3$ and brown (Fe, Mn)$_2O_3$ particles was not needed because it would have lowered the point of the zero-charge value at a 2.0 pH even more. The zeta-potential dependence on the amount of titrant (ammonium citrate) was measured only for the black $Fe_3O_4$ and $\alpha$-FeOOH yellow-particle solutions (Figure S1b). Table S1 summarizes the absolute density, specific surface, and calculated BET average particle sizes for the best-performing red $Al_2O_3$: $Fe_2O_3$ slurry compounds; Figure S2 shows the volume-based LD particle size distributions of the red $Al_2O_3$: $Fe_2O_3$ ceramic slurry components.

To achieve a uniform zirconia-based coating, a bimodal mixture of iron oxide $\alpha$-$Fe_2O_3$ and $ZrO_2$ nanoparticles was used. Both alumina-based and zirconia-based coatings were spin-coated with a 1500 rpm speed for 15 s on various substrates: on 1 mm-thick soda-lime glass to achieve 3 $\mu$m-thick (alumina) and on 30 $\mu$m-thick flexible stainless-steel foil to obtain 200 nm-thick (zirconia) coatings. The slurry solution was dried at 90 °C on a hot plate for 5 min after spin coating.

**FLA and temperature-profile simulation.** The FLA of the oxide ceramic alumina films doped with 5 vol% of $Fe_xO_y$ red/brown/black/yellow pigments was performed in air with a photonic curing system (NovaCentrix PulseForge 1300). For the FLA process, samples were positioned 10 mm away from a Xe arc lamp, with the ceramic film side directed toward the incident light. One pulse had a 2500 $\mu$s envelope comprising five 400 $\mu$s long micropulses, with a 100 $\mu$s break after each micropulse. Repeating the FLA treatment led to the enlarging of the sintered film area. Therefore, consequent FLA pulses, using an 850 V lamp voltage with five-pulse repetition (with a total output exposure energy density of 125 J/cm$^2$), were applied. SimPulse software was used to simulate the maximum temperature reached on the ceramic coating surface, as well as the temperature at the coating/substrate interface. For the simulations, the materials stacked 1 mm-thick SLG and 3 $\mu$m-thick alumina, or 200 nm zirconia coating. The correction of the coating absorptivity due to the presence of $Fe_xO_y$ red/brown/black/yellow pigments was made according to each color case for alumina, and due to the $Fe_2O_3$ red pigments for zirconia.

**Characterization methods.** The morphologies and thicknesses of the layers, as well as the film compositions, were evaluated by scanning electron microscopy (SEM) and EDX on an FEI Quanta 650 SEM. Transmittance was measured with a UV–Vis spectrometer (Shimadzu UV-3600) from 250 to 1500 nm, taking air as the reference (baseline). X-ray diffraction patterns were measured on an X'Pert Pro in Bragg–Brentano geometry using Cu K$\alpha$1 radiation ($\lambda$ = 1.5406 Å), scanning from 20° to 80° (2$\theta$), with a step interval of 0.0167°. The size distribution of the dispersed particles in the water-based slurry was measured by laser diffraction (LS 13320, Beckman Coulter GmbH, Krefeld, Germany). The absolute densities of all the powders were measured by helium pycnometry (AccuPyc II 1340, Micromeritics, USA). BET measurements (SA 3100 Surface Area Analyzer, Beckman Coulter GmbH, Krefeld, Germany) allowed us to calculate the specific surface areas and BET average sizes of the particles.

## 3. Results and Discussion

Figure 1 schematically describes the structures of the ceramic precursor and iron oxide pigments, with corresponding SEM images. To obtain thin and uniform layers, we utilized a bimodal mixture of the $\alpha$-$Al_2O_3$, a submicron $\alpha$-$Al_2O_3$ doping of red/brown/black/yellow $Fe_xO_y$ pigments in DI water, as described in the "Materials and methods" section. The powdered additives (Figure 1a) have intense colors, which are aimed to increase the overall absorption of the white-colored alumina precursor powder at the applied wavelength range

(400–800 nm). A homogenous three-modal distribution was obtained with the nanometer $Al_2O_3$ and $Fe_xO_y$ additives that filled the gaps between the micrometer $Al_2O_3$ particles (Figure 1b). SEM images of the black/yellow/brown and red $Fe_xO_y$ pigments (Figure 1c) demonstrate that red $\alpha$-$Fe_2O_3$, black $Fe_3O_4$, and brown (Fe, Mn)$_2$O$_3$ nanoparticles have a spherical shape, with average sizes of 90, 150, and 300 nm, respectively (material data specification), while yellow $\alpha$-FeOOH particles, with a 200 nm $\times$ 300 nm size, have a prismatic elongated shape. Among the transition metals (Cr, Fe, Eu, Yb, Dy, Tb), the Fe dopant not only enhances sintering, but also improves the catalytic, magnetic, and optical properties of alumina due to the high surface density and the ion radius close to the Al [24,25]. By adding impurities, $Fe^{3+}$ ions enter the structure of alumina and significantly reduce the bandgap of the material.

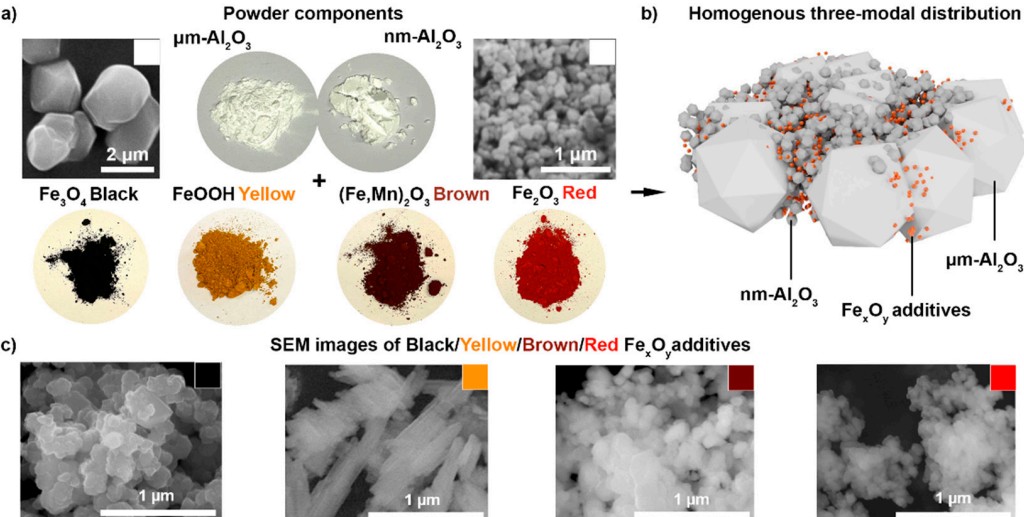

**Figure 1.** The structures of the ceramic precursor and colored iron oxide particles: (**a**) powder components comprising a bimodal mixture of micrometer-sized $\alpha$-$Al_2O_3$ and nanometer-sized $\alpha$-$Al_2O_3$, with corresponding SEM images, mixed with one of the four pigments, which resulted in (**b**) a homogeneous three-modal mixture of alumina particles and $Fe_xO_y$ additives; (**c**) SEM images of the black/yellow/brown and red $Fe_xO_y$ additives showing their sizes/shapes and morphologies.

Water-based slurries with four different-colored (red $\alpha$-$Fe_2O_3$, black $Fe_3O_4$, brown (Fe, Mn)$_2$O$_3$, and yellow $\alpha$-FeOOH) additives were prepared with the use of ammonia citrate dibasic, as shown in Figure 2a. The spin-coating of these slurries resulted in colored coatings on glass with different color shades and intensities (Figure 2b). From the photographs, the most intense color is observed for the red and brown pigments, while yellow coatings, and especially black coatings, exhibit low coloration. It was proven that the powder properties, such as the transparency, particle size, and concentration, are the most important factors that affect the color intensity of the powder [26]. Thus, having the same vol% of all four pigments, the smaller particles (red: 90 nm and brown: 150 nm) provide the highest color intensity. Due to the homogeneity of the spin-coated layers on glass, it was possible to characterize the optical properties. Because the emission spectrum of the xenon lamp (Figure 2c) has a broad peak in the visible region (400–800 nm), the absorption obtained by the $Fe_xO_y$ pigment was sufficient for the subsequent FLA. It can be seen in Figure 2c that the highest absorption in the peak region of the FLA lamp spectra (from 400 to 500 nm) was obtained in the case of the red $\alpha$-$Fe_2O_3$ pigment, which was in the range of 88%, while the absorption was only 60%, 41%, and 22% for the brown-, yellow-, and black-colored layers, respectively. The colored pigments play the key role in the sintering process of ceramic-based films by FLA because these particles are necessary to generate enough light absorption, which is followed by a rapid heat transfer from the $Fe_xO_y$ pigments to the

base alumina particles (Figure 2d). At this point, it was expected that the highest sintering activity would be achieved for the red $Al_2O_3$: $Fe_2O_3$ coating.

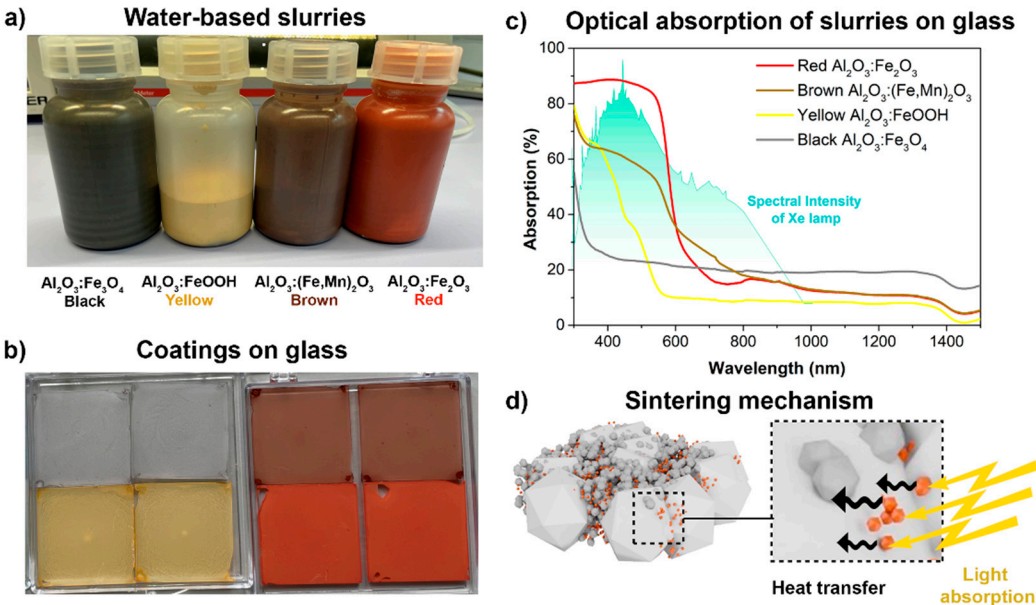

**Figure 2.** Characterization of slurries and precursor coatings on glass: (**a**) image of the water-based slurries with four different colors, with red $\alpha$-$Fe_2O_3$, black $Fe_3O_4$, brown $(Fe, Mn)_2O_3$, and yellow $\alpha$-FeOOH pigments; (**b**) corresponding images of the slurries with various color pigments spin-coated on glass; (**c**) optical absorption spectra of as-coated precursor films on glass superimposed with the FLA lamp spectra; (**d**) schematics of the ceramic sintering mechanism showing light absorption by the colored iron oxide pigments, followed by heat transfer to the matrix e alumina particles.

After the spin-coating and drying of the colored precursor films, FLA was performed in air. The temperature reached in the films upon the given FLA parameters was simulated by using SimPulse software (Figure 3a). According to the simulated temperature profile (Figure 3a), a maximum annealing temperature of 1850 °C was reached for the red $Al_2O_3$: $Fe_2O_3$ ceramic film, while maximum temperatures below 1200 °C, 500 °C, and 400 °C were reached for the brown $Al_2O_3$: $(Fe, Mn)_2O_3$, yellow $Al_2O_3$: FeOOH, and black $Al_2O_3$: $Fe_3O_4$ ceramic films, respectively. The SEM images (Figure 3b) illustrate the morphology changes in the films, which occurred as a result of the photonic sintering. Only the red $Al_2O_3$: $Fe_2O_3$ ceramic film experienced a final-stage sintering, and the morphology of the homogeneous dispersed initial mixture of the micrometer and nanometer-sized $\alpha$-$Al_2O_3$ particles with various pigments changed to a sintered-grain structure. In the case of brown $Al_2O_3$: $(Fe, Mn)_2O_3$ particles, only the initiation of sintering with a starting sintering neck formation was observed. For the black $Al_2O_3$: $Fe_3O_4$ and yellow $Al_2O_3$: $\alpha$-FeOOH particles, no sintering was detected, which can be explained by the lower optical absorption, which was confirmed by the optical measurements of the slurries on glass and, therefore, the lower temperatures reached on the surface.

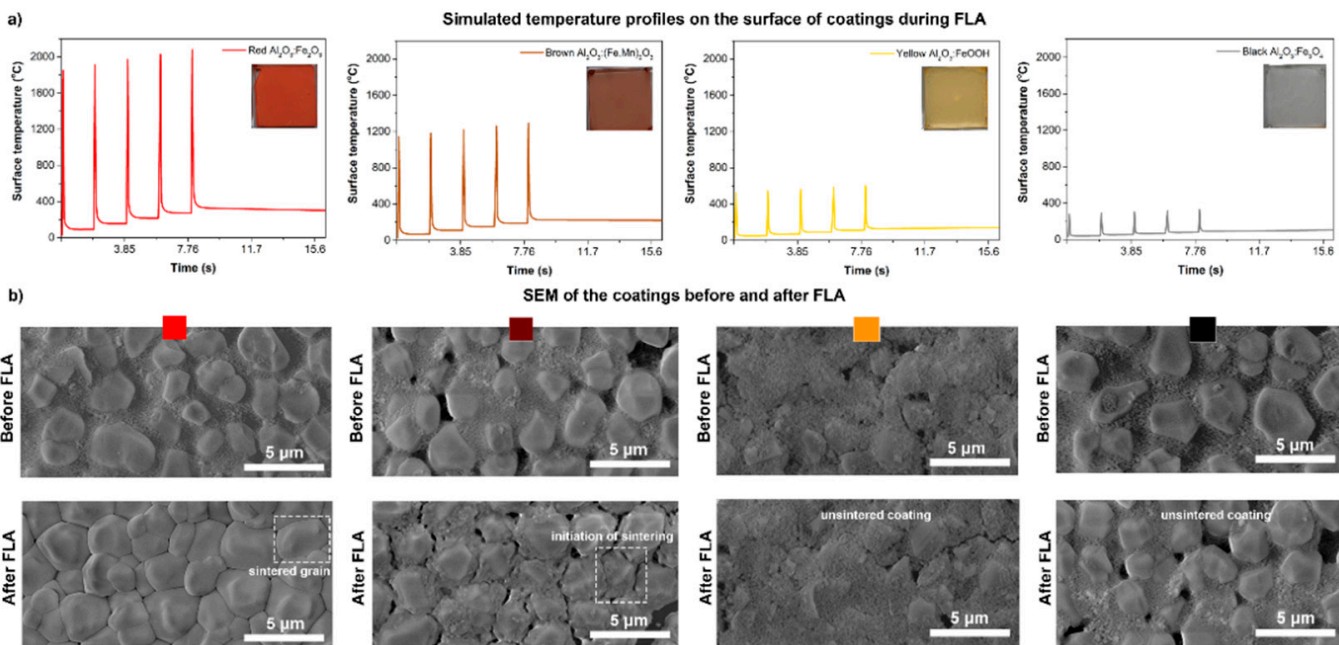

**Figure 3.** Compositional analysis of the ceramic films before and after FLA: (**a**) simulated temperature profiles on the surfaces of different films obtained using SimPulse software; (**b**) top-view SEM images of the ceramic films before and after FLA for various pigments, demonstrating that the most pronounced sintering occurred for the red $\alpha$-$Fe_2O_3$ pigment.

We focus on the red $Al_2O_3$: $Fe_2O_3$ layers because it was possible to obtain uniformly sintered coatings in this case. The BET specific surface area and density of Bayferrox Red 110 ($\alpha$-$Fe_2O_3$) are 15.2 $m^2$/g and 5.0 g/$cm^3$, respectively, from the data specification, and so the calculated BET average particle diameter is 104 nm. The phase compositions of these films were analyzed by XRD in a grazing incidence mode before and after FLA (Figure 4a). Both patterns match well with the diffraction pattern of the trigonal $\alpha$-$Al_2O_3$: reflexes at 26, 35, 38, 43, 52, 57, 61, 66, 68, and 77° correspond to the reference pattern ICSD 52648. The reflexes at approximately 24, 33, and 41 and 50, 54, and 62° (marked with red arrows) for the unprocessed layer correspond to $\alpha$-$Fe_2O_3$ and are not present in the patterns collected from the FLA-processed layer, which indicates the melting of the light-absorbing iron oxide. Both corundum ($\alpha$-$Al_2O_3$) and hematite ($Fe_2O_3$) belong to the same space group (R$\bar{3}$c), with lattice parameters of a = 4.760 Å and c = 12.993 Å, and a = 5.039 Å and c = 13.740 Å, respectively, and they can form solid solutions. It can be observed that all the main peak positions slightly shifted to a lower 2θ degree (by 1°) from Figure S3, which indicates the cell-volume increase. The obtained results are in accordance with the abovementioned crystallographic data and can be explained by the implementation of Fe ions within the corundum lattice [23], which can also be seen from the EDX elemental map collected from the sample after FLA (Figure S5). In addition, it can be observed that the main $Al_2O_3$ peaks at 26, 35, 38, 43, 52, 57, and 68° have higher peak intensities (maximum peak heights), as well as lower FHWM values (narrow peak shapes) for the pattern obtained after FLA, compared with the one before the FLA. This may be due to the crystallite shape and size change caused by the FLA—from bimodal alumina particles before the FLA to the sintered alumina grains [27]. Figure S4 additionally shows the Al 2p, Fe 2p, and O 1s XPS peaks of the red $Al_2O_3$: $Fe_2O_3$ layer samples before and after the FLA. The results show that the Al 2p peak was at 71.8 eV in the case of $Al_2O_3$: $Fe_2O_3$ before FLA, whereas it was shifted to a lower binding energy (71.0 eV) for the sample sintered by FLA; the same trend was observed for the O 1s peak. The shift in the Al 2p peak towards a lower binding energy after the FLA is mainly attributed to the decrease in the coordination number and amount of $Al^{3+}$ ions in the film. Similar results were observed for $AlO_x$ thin films annealed at higher

temperatures [28]. Various SEM images (Figure 4b) confirm the formation of the well-defined grains, which reach a size of 1–5 μm after grain growth via sintering. The $\alpha$-Fe$_2$O$_3$ nanoparticles are assumed to initiate grain growth by local melting after FLA due to their effective light absorption and heat conduction. It can also be seen from the cross-sectional SEM that the grains are merged between each other, with no gaps or pinholes.

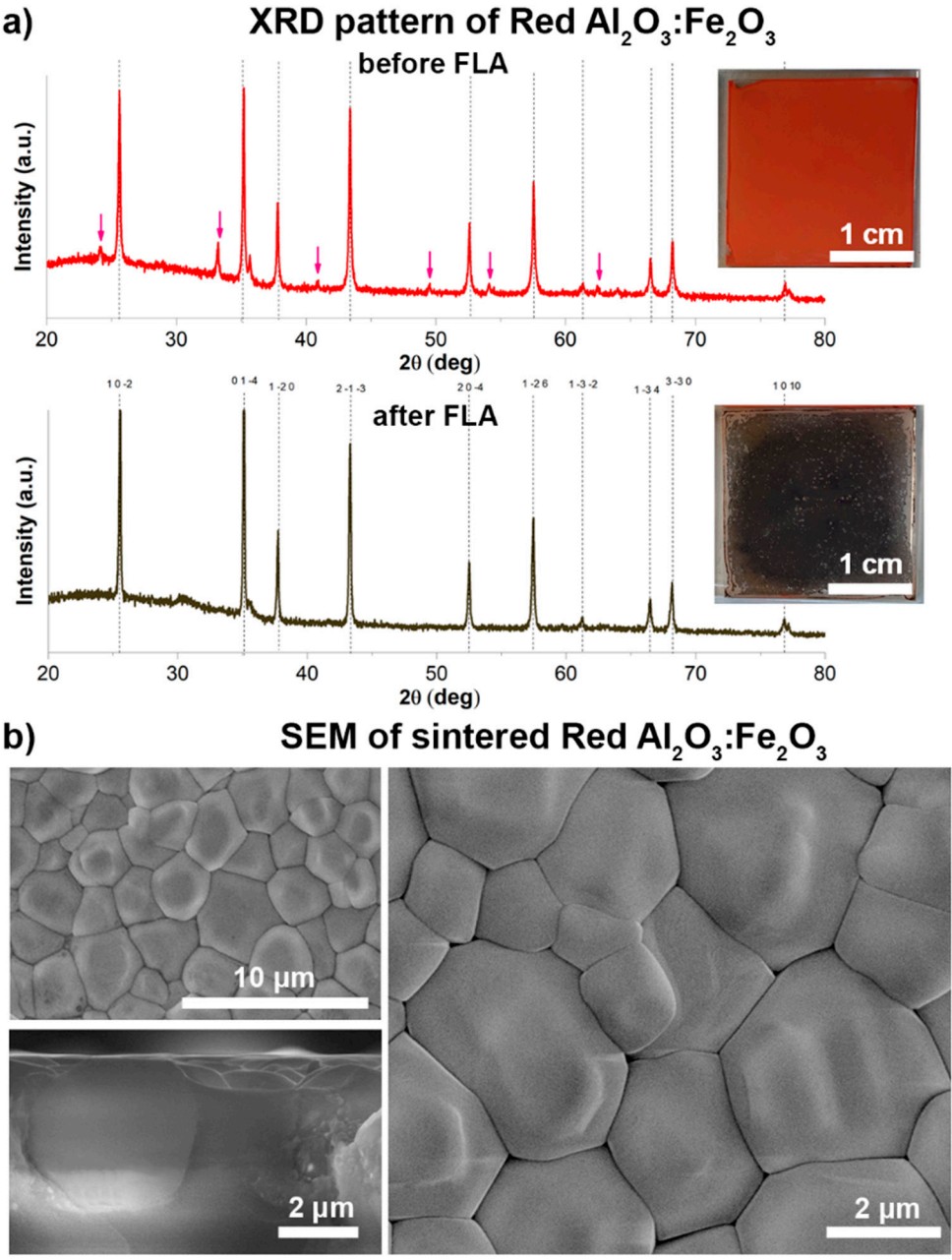

**Figure 4.** Characterization of the red Al$_2$O$_3$: Fe$_2$O$_3$ ceramic film before and after FLA: (**a**) indexed XRD patterns of the ceramic films with representative sample photographs (red arrows showing the $\alpha$-Fe$_2$O$_3$ pattern); (**b**) top-view SEM images of FLA-sintered red Al$_2$O$_3$: Fe$_2$O$_3$ ceramic films with different magnifications, as well as cross-sectional SEM image of the sintered film.

To check whether the selected red a-Fe$_2$O$_3$ pigments can be employed for the FLA of other refractory oxide ceramics, a thin film of a red ZrO$_2$: Fe$_2$O$_3$ bimodal mixture composed of nanometer-sized ZrO$_2$ and $\alpha$-Fe$_2$O$_3$ additives (Figure 5a) was deposited onto a glass slide. The SEM morphological characterization of the layer before FLA confirms a homogeneous distribution of the nanometer-sized particles in the layer, with a thickness of

around 200 nm (Figure 5b). The optical absorption spectrum of the as-prepared red ceramic film exhibits a high absorption of the Xe-lamp spectra: from 40 to 70% in a 400–550 nm wavelength range (Figure 5c). The red $ZrO_2$: $Fe_2O_3$ coating contained much pigment, and the layer thickness was only 200 nm, and so it was not possible to sinter the layers on glass because they were destroyed during the FLA. The temperature reached at the interface of the coating and the glass was simulated to be more than 900 °C. The low softening point of the glass, as well as the thermal-expansion mismatch with the ceramic layer, resulted in the unfortunate damaging and delaminating of the top layer of the glass due to the heat accumulation at this interface. Therefore, we changed the substrate to stainless steel for the FLA-assisted sintering of the thin red $ZrO_2$: $Fe_2O_3$ coating. The stainless steel has a higher thermal conductivity and is therefore able to withstand a high temperature at the interface. Simulated temperature profiles on the surface of the ceramic layer during the FLA process confirmed the high temperature reached by the absorption onto the flexible stainless-steel foil (thickness of only 30 μm) (Figure 5d). A zirconia-based coating was sintered with FLA (Figure 5e). Figure S6 contains the XRD patterns of the red $ZrO_2$: $Fe_2O_3$ ceramic film before and after FLA. In the initial state, when pure $ZrO_2$ was mixed with $Fe_2O_3$, the highly crystalline monoclinic phase was observed. The phase transition of $ZrO_2$, and the appearance of the tetragonal phase dominating over the monoclinic, have been observed after FLA. This effect was reported previously for a series of samples annealed at temperatures up to 1100 °C [29]. The XPS measurements of both red $ZrO_2$:$Fe_2O_3$ layers before and after FLA are shown in Figure S7. The Zr $3d_{5/2}$ binding energies of tetragonal (dominated after FLA) and monoclinic (before FLA) films differ by 3.6 eV. In the O 1s region, this difference is 3.1 eV. The Zr 3d and O 1s binding energies depend on the exact preparation parameters (i.e., temperatures during deposition and annealing, and film thickness). It was demonstrated earlier that, by annealing at 920 °C, the film can be transformed into the tetragonal structure, which is attributed to the reduction in the film (i.e., the formation of oxygen vacancies, which stabilize the tetragonal phase [30]). The transformation of the monoclinic phase of $ZrO_2$ to tetragonal broadens its applications in devices such as solid oxide fuel cells, which operate over a broad temperature range.

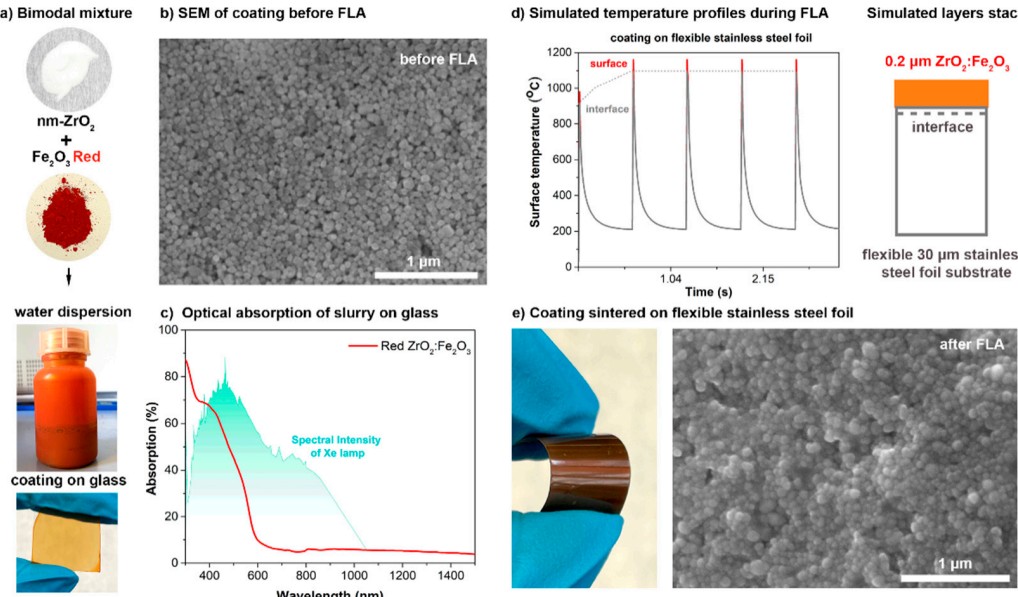

**Figure 5.** Characterization of the red $ZrO_2$: $Fe_2O_3$ ceramic film before and after FLA: (**a**) powder components comprised a bimodal mixture of nanometer-sized $ZrO_2$ and nanometer-sized red α-$Fe_2O_3$, which were mixed with a water-based slurry and deposited onto a glass slide for the (**b**) SEM morphological characterization and (**c**) optical absorption spectra of as-prepared red ceramic films; (**d**) simulated temperature profiles on the surface of the ceramic layer during the FLA process for layer deposited on glass and (**e**) flexible metal foil.

## 4. Conclusions

We have investigated the effect of colored iron oxide pigments on the FLA of oxide ceramic coatings. The precursor layers comprised white ceramic particles of alumina or zirconia, mixed with 5 vol% of red ($\alpha$-Fe$_2$O$_3$), brown ((Fe, Mn)$_2$O$_3$), yellow ($\alpha$-FeOOH), and black (Fe$_3$O$_4$) nanoparticle pigments. We have identified the key criteria for the identification of a coloring agent that is suitable for the FLA processing of ceramics: the highest absorption in the peak region of the FLA lamp spectra (from 400 to 550 nm), as well as the smallest particle size. The highest absorption (in the range of 88%) was obtained for the alumina layer colored by red 90 nm-sized $\alpha$-Fe$_2$O$_3$ particles, whereas the optical absorptions were lower than 60% for the brown, yellow, and black pigments. The FLA process resulted in sintered thin alumina ceramic films on a glass substrate in just 7 s for the red-colored alumina precursors. Such coatings can potentially improve the corrosion resistance of metals, such as steel, as well as prevent or minimize the decarburization of the steel surface during exposure at elevated temperatures. We have extended the proposed approach of adding coloring pigments into a zirconia-based slurry, obtaining 200 nm sintered coatings on a flexible metal-foil substrate. Due to the biocompatibility of zirconia, such coatings can be used as bioceramics for bone implants, the activator of the calcium phosphate-complex growth on the surface of metallic implants being a naturally strong adhesive to human bones. This general approach can therefore be extended for the photonic sintering of any oxide-based ceramic film.

**Supplementary Materials:** The following supporting information can be downloaded at: https://www.mdpi.com/article/10.3390/ceramics5030027/s1, Figure S1: zeta potential a) of all colored Fe$_x$O$_y$ nanoparticle pigments in water, b) dependence on the amount of added titrant (ammonium citrate); Table S1: Absolute density and specific surface area (SSA) of raw powders measured by helium pycnomerty and BET measurements, respectively, and calculated BET average particle sizes for all powders; Figure S2: Volume-based LD particle size distribution of Red Al$_2$O$_3$:Fe$_2$O$_3$ ceramic slurry in water dispersed by means of ammonium citrate. The first broad peak corresponds to the mixture of nm-Al$_2$O$_3$ and nm-Fe$_2$O$_3$, while the second peak—to the μm-Al$_2$O$_3$; Figure S3: Enlarged XRD pattern peaks (from Figure 4a)—(1 0 -2), (0 1 -4), (1 -2 0), and (2 -1 -3)—indicating the small shift of the corundum peaks by 1° 2θ; Figure S4: Al 2p, Fe 2p, and O 1s XPS peaks of the Red Al$_2$O$_3$:Fe$_2$O$_3$ layers samples before and after FLA. Al and O peaks indicate a shift towards lower binding energies for the samples after FLA; Figure S5: EDX maps of the grains interconnection for the FLA-sintered Red Al$_2$O$_3$:Fe$_2$O$_3$ layer; Figure S6: Indexed XRD patterns of the ceramic Red ZrO$_2$:Fe$_2$O$_3$ ceramic film before and after FLA; Figure S7: Zr 3d, Fe 2p, and O 1s XPS peaks of the Red ZrO$_2$:Fe$_2$O$_3$ layers samples before and after FLA. Zr and O peaks indicate a shift towards higher binding energies for the samples after FLA.

**Author Contributions:** The conceptualization of the study was conducted by E.G. and S.P. The experimental measurements were conducted by E.G. and S.S. The analysis and interpretation of the results, as well as the conclusions, were conducted by all the co-authors. The manuscript was written by E.G., with the revision and approval of the other co-authors. Y.E.R., V.V. and T.G. participated in the discussions and helped to revise it as supervisors. All authors have read and agreed to the published version of the manuscript.

**Funding:** The EMPAPOSTDOCS-II programme received funding from the European Union's Horizon 2020 research and innovation programme under the Marie Skłodowska-Curie Grant, Agreement Number 754364. An exceptional thankyou goes to the SFAs (Strategic Focus Areas) Advanced Manufacturing Program 2021–2024: "Functional Integration for rapid realization of microreactors and bio-assays".

**Institutional Review Board Statement:** Not applicable.

**Informed Consent Statement:** Not applicable.

**Data Availability Statement:** Not applicable.

**Conflicts of Interest:** The authors declare no conflict of interest.

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
