# Peer review of "Photonic Sintering of Oxide Ceramic Films: Effect of Colored FexOy Nanoparticle Pigments"

_ceramics, doi:10.3390/ceramics5030027_

Round 1

Reviewer 1 Report

This work is devoted to the production of alumina and zirconia thin films modified with colored FexOy nanopigments. Such films were obtained by flash lamp annealing.

The characteristics of the thin films obtained with different additives were compared, including surface morphology, optical properties, crystallinity, and structure.

The work can be accepted for printing in Ceramix, but requires the following corrections and refinements:

1. The work lacks the results with particle size distribution to show what the initial raw material for thin films consists of. From one SEM picture it is hard to tell how homogeneous the powders

2. It is desirable to apply the RGB color characteristic to understand the hue

3. Whether XRD studies have been evaluated to understand the formation of solid solutions in films

4. It is interesting to compare the effect of commercial powders with synthesized powders. What technologies for producing powders are currently known wet synthesis and dry synthesis please complete the introduction

5. It is desirable to evaluate the effect of the porosity of the obtained films using BET gas adsorption methods 10.1016/j.matchar.2018.08.044

Author Response

We have addressed all the points highlighted by the Reviewer in the file attached below.

Reviewer 2 Report

It would be good to tell where and why thin ceramic films are used. In this regard, it is important in the discussion of the results to touch on the effect of pigments on the quality of the obtained ceramic films. How appropriate is the use of pigments.

Author Response

(The authors gave the same response as above.)

Reviewer 3 Report

The current work focuses on Photonic sintering of oxide ceramic films: effect of colored FexOy nanoparticle pigments. The author’s great effort into the manuscript. Minor issues should address.

Abstract

The first appearance of the abbreviation should have a full definition e.g. FLA

Introduction

- The introduction is general and does not provide sufficient background, and all relevant references are not included 

- In addition, the novelty of this work is not highlighted and it was not clear the author's contribution in comparison to other previous works.

Experiment section 

-Line 68-78, it should under heading materials

- The heading for Particle dispersion start from Line 79-95

- Experimental part required rephrasing to be more precise with more details for the reader

Result and discussion 

- Line 120-124 seems to belong to the experimental section, please remove them from the results section

- The EDX elemental map is required to show the film's composition and the Fe localization at the grain boundaries.

- Why use this ratio of only “5 vol% of FexOy red/brown/black/yellow pigments”

- Fig 1c, the SEM picture should replace with another high-quality and good resolution to distinguish the morphology and particle size distribution 

- It stat “The water-based slurries with four different colors, α-Fe2O3 Red, Fe3O4 Black, 133 (Fe,Mn)2O3 Brown and α-FeOOH Yellow additives were prepared with the use of ammonia citrate dibasic” 

No information in the experimental section on how these water-based slurries were prepared? or what amount of ammonia citrate dibasic is used? More details are required in the experimental section for easier reproducible by the reader

- Label on fig 2c to clear which materials it belong of this coating on glass

- It stat “From the photographs it is visible that the most intense color is observed for red and brown pigments, while yellow, and especially black coatings exhibit a low coloration. This can be due to the different sizes of the additives (red: 90 nm; brown: 150 nm, black and yellow: up to 300 nm) resulting in a lower optical color intensity due to a lower amount of particles, despite having the same vol% of pigment.”

No information about the relation between the size and the color. Please rephrase this paragraph to clear it. Also, citations should insert to confirm this claiming.

- Indexed peak should insert in Fig 4a easier comparing.

- One of the main problems in the manuscript is that the authors show only results without interpretations of it or confirmation by citation. More details are required to explain the obtained results.

References

- References should have one system with complete citation e.g. ref. 12 and ref. 18

Author Response

(The authors gave the same response as above.)
